# Accurate and Rapid Genetic Tracing the Authenticity of Floral Originated Honey with the Molecular Lateral Flow Strip

**DOI:** 10.3390/bios12110971

**Published:** 2022-11-04

**Authors:** Qian Wu, Qi Chen, Chao Yan, Jianguo Xu, Zhaoran Chen, Li Yao, Jianfeng Lu, Bangben Yao, Wei Chen

**Affiliations:** 1Engineering Research Center of Bio-Process, MOE, School of Food and Biological Engineering, Hefei University of Technology, Hefei 230009, China; 2Product Quality Supervision and Inspection Research Institute of Anhui Province, Hefei 230009, China; 3Intelligent Manufacturing Institute, Hefei University of Technology, Hefei 230009, China

**Keywords:** locust honey, authentication, molecular amplification, lateral flow strip, food safety

## Abstract

Honey is a natural product and is heavily consumed for its well-known nutritional functions. Honeys with different floral origins possess distinctive flavors, tastes, functions and economic values. It is vital to establish an effective strategy for identifying the authenticity of honey. The intrinsic genetic materials of pollen were adopted as target analytes for the effective identification of honey with floral origins. With an optimized protocol for the rapid gene extraction from honey, target genetic templates were amplified on-site with a portable device. Conveniently, all on-site amplified functional products were easily judged by the designed lateral flow strip (LFS), which was defined as the molecular LFS in this research. Additionally, the entire on-site genetic authentication of honey was completed in less than 2 h by visual observation. Commercial honey products have been successfully identified with excellent accuracy. This low-cost, high-efficiency and easy-operational strategy will greatly benefit the quality guarantee of foods with specific functions and geographical markers.

## 1. Introduction

Honey—a natural, sweet substance produced by bees—is a mixture of collected nectar, the secretions or honeydew from plants and the secretions from bees. Honey is rich in essential amino acids, vitamins, polyphenols, flavonoids, polysaccharides, active enzymes, carotenoids and other active substances. These components perform physiological functions including antioxidation, antibacterial support, the acceleration of wound repair, the enhancement of immunity et al. [1], and benefit the health of consumers. According to the Commission of the European Communities, honey that is sold must indicate its floral origins, physicochemical properties and organoleptic characteristics [2,3]. Among these labeling items, physicochemical properties can be easily determined by various analytical methods, unlike the floral origins of honey. 

However, the quality of honey is highly and strongly dependent on its floral origins, which also determines the honey’s organoleptic properties and nutritional effects [4,5,6]. For example, one of China’s popular categories of honey, the locust honey, is produced from the pollen of locust trees, and demonstrates the functions of: lowering the lipid level and pressure of blood, preventing apoplexy, maintaining the smoothness and glossiness of the skin and regulating the emotions of consumers [7]. These demonstrated functions can be ascribed to special components in the acacia flowers. Meanwhile, it is also common knowledge that monofloral honey originating from distinct single-plant species is usually distributed with a higher price than that of honey originating from mixed botanical species [5,8]. Moreover, allergens in honeys with mixed components can lead to serious threats to both the mental health and the quality of daily life in specific populations [9,10]. Therefore, in order to avoid the business fraud induced by unfair trading, and to guarantee the health of consumers, the precise identification of honey purity is a priority for both producers and regulatory authorities [11,12]. Traditionally, numerous analytical methods have already been employed for the authentication of honey components including thin-layer chromatography (TLC), gas chromatography-mass spectrum (GC-MS), high-performance liquid chromatography (HPLC), near-infrared spectroscopy (NIR), nuclear magnetic resonance (NMR), high-performance anion exchange chromatography (HPAEC) and so on [12,13]. These instrumental-based methods have contributed greatly to the regulation of the honey market. However, each of the afore-mentioned instrumental methods may only be performed by professional operators in the central labs; additionally, certain illegal producers are familiar with the previously proven protocols, and the advanced adulteration of honey can effectively escape supervision. Intrinsically, regardless of their origination from mixed botanicals or from syrups, the adulterated components may be well differentiated using genetical strategies [8,14,15,16]. The afore-mentioned genetic amplification methods, which use expensive and immovable devices, should also be conducted by technicians with a professional background [17,18]. Significantly, the detection results—which usually regard the fluorescent signal-based Ct curves—may only be judged and explained by professional operators, which also inhibits wide applications in manufactory or on-site detection [19,20]. The means with which to further energize a powerful amplification strategy for the rapid and simple identification of the authenticity of floral originated honey continues to be much needed for mass consumers. Considering the simplicity and the easy judgement of its final results, the lateral flow strip (LFS) has been widely used in food safety, clinical diagnosis and pandemic control since the first commercialization of the early pregnancy diagnosis kit [21,22]. This research is inspired by these practical demands for the identification of honey; the rapid on-site treatment of honey samples, on-site amplification and visual judgement of the identification results of honey’s floral origins have been systematically performed and optimized. Rapid genetic tracing of the authenticity of target honey has been well achieved with excellent simplicity and efficiency, which has greatly facilitated the routine screenings or identifications of the authenticity of honey.

## 2. Materials and Methods

### 2.1. Chemicals and Reagents

Casein, Casein-Na, and bovine serum albumin (BSA) were obtained from Sigma-Aldrich (Shanghai, China). All oligonucleotides—including the forward primer 5′-biotin-GCTCAACCAGGAACGATC-3′ and reverse primer 5′-FITC-GCAGCATTTGACTACGTACCA-3′ (with expected amplified length of 112 bp); 4S Red Plus nucleic acid stain (1000×); Taq DNA polymerase (5 U/μL); 10 × PBS; streptavidin; agarose; dNTP mix (25 mM); and proteinase K solution (20 mg/mL)—were all obtained from Sangon Biotech (Shanghai, China). A S1000 Thermal Cycler PCR (Bio-Rad, Hercules, CA, USA) and the portable thermal controller (Hangzhou Ao-Min Biological Co., Ltd., Hangzhou, China) were used for the amplification. All components of the LFS, including a plastic adhesive backing, sample pad, conjugate pad, nitrocellulose membrane CN 95 and absorbent pad were ordered from the Shanghai Jie-ning Biotechnology Co., Ltd. (Shanghai, China). All solvents and other chemicals of analytical reagent grade were used without further purification.

### 2.2. Rapid Pre-Treatment of Honey Samples for Genetic Tracing of Authenticity

In order to realize the rapid genetic tracing of honey, the pretreatment of honey samples with a highly viscous property is of critical significance. All honey samples were provided and confirmed by the Product Quality Supervision and Inspection Research Institute of Anhui Province, China. For the verification of the general applicability of the treatment protocol, different honey samples (locust honey, rape honey, hazelnut honey, chaste honey and jujube honey) and related mixtures were all treated for analysis. Additionally, 14 commercial honey samples, purchased from the local supermarket, were also analyzed with the method developed in this research. Typically, 500 μL of the different treated honey samples, 700 μL of lysis buffer (10 mM Tris-HCl, 200 mM NaCl, 5 mM EDTA, 1% SDS, pH 8.0) and 30 μL of Proteinase K were mixed vigorously for 30 s and sonicated for 5 min. Following this, the mixture was centrifuged at 12,000 *g* for 10 min and the supernatant was transferred to a new tube. 10 μg carboxylated magnetic nanoparticles (MNPs) and 350 μL of coating buffer (2 M NaCl and 30% PEG 20,000) were mixed with the supernatant for 5 min to adsorb the target DNA. The MNPs with absorbed DNA were finally collected via magnetic separation and washed twice with 70% ethanol to remove other impurities. The extracted genetic materials were finally eluted and dispersed in 25 μL TE solution (10 mM Tris-HCl, 1 mM EDTA, pH 8.0) and confirmed by UV-vis spectrophotometry prior to use. The collected DNA was stored at −20 °C for later use.

### 2.3. Preparation of GNPs and GNP-Labeled Anti-FITC Antibody Conjugates

The gold nanoparticles (GNPs) utilized in this research were prepared based on the classic citrate reduction method. Firstly, the flask was washed and cleaned thoroughly with aqua regia solution and double-distilled water, respectively. An amount of 100 mL chloroauric acid solution (0.1 g/L) was added to the 250 mL flask with magnetic stirring and was heated to boiling point. Afterward, 1% trisodium citrate solution was rapidly injected at a certain volume while stirring continued at 1500 r/min. The solution underwent a significant color change after one minute. Typically, the color of the solution changed from light yellow to gray, then to black, then to purple; finally, the color remained a stable wine-red. The final solution continued to be stirred for another 5 min. The obtained GNPs were stored at 4 °C for subsequent labeling research. Typically, 5 μL 0.1 M K_2_CO_3_ and 3 μL anti-FITC antibody were added into 500 μL above the prepared GNPs and gently shaken for 1 h. The residual active sites on GNPs were blocked by adding 50 μL BSA (10%), reacting for 30 min. Excess antibodies and other reagents were removed by centrifugation at 9000 rpm for 10 min. The remaining precipitates were dissolved in 50 μL BSA-contained buffer and sprayed to prepare the conjugation pad for the assembly of the LFS. 

### 2.4. Assembly of LFS for Rapid Analysis of Functional Amplicons

For the assembly of LFS for the rapid analysis of functional amplicons, the test line and control line of the LFS were first fabricated. Streptavidin (1 mg/mL) and the secondary antibody (1 mg/mL) were sprayed onto the NC membrane at a rate of 0.5 µL/cm as the T line and C line, respectively. For preparation of the conjugation pad, 4 μL of GNP-FITC-antibody conjugate was dropped onto the glass fiber and dried in a constant temperature oven at 30 °C for 4 h. The sample pad, conjugate pad, NC membrane and absorbent pad were then sequentially assembled on a single plastic adhesive backing pad. Lastly, the prepared lateral flow strip was cut into small pieces with a width of 3 mm and stored in dry and dark conditions.

### 2.5. On-Site Rapid Production of Functional Amplicons of Honey for LFS Analysis

The on-site production of functional amplicons was performed on a portable thermal controller with the volume of 25 µL containing: 12.5 µL 2 × PCR Mix buffer (2.0 mM MgCl_2_, 200 µM of dNTPs and 5 U Taq DNA polymerase); 1 µL functional forward primer (10 μM) and reverse primer (10 μΜ); 1µL DNA templates; and 10.5 μL ultrapure water. All amplifications were conducted based on the following program: denaturation at 95 °C for 5 min, followed by 30 cycles of 94 °C for 20 s, 53 °C for 20 s, and 72 °C for 30 s. A final extension was performed at 72 °C for 5 min. Followingly, 1 µL obtained functional amplicons were dropped onto the sample pad and 50 µL PBS (10 mM) was further added to provide the migration force on the LFS. The results can be judged by observation after 3 min for adulterated honey, as opposed to after 5 min for non-adulteration judgement. For the genetic authentication of commercial honey products with the molecular LFS, all the honey samples can be directly and rapidly treated as described above.

## 3. Results and Discussion

### 3.1. Working Principle of the Rapid Genetic Authentication Strategy with Molecular LFS

For the rapid identification of the floral origin of honey, the genetic authentication strategy with the molecular LFS is designed and demonstrated in Figure 1. The optimized rapid pretreatment of honey samples can be completed by ultrasonication and magnetic extraction. With the ingeniously designed primer set functionalized with biotin and FITC, respectively, the sequence of target floral origin can be well and on-site amplified with the portable thermal controller in 50 min; this will produce an enormous number of dual-labeled functional amplicons. When the dual-labeled amplicons were loaded onto the LFS, the dual-labeled amplicons will firstly be recognized by the Anti–FITC–GNP via specific antigen−antibody interaction on the conjugate pad, and then be trapped by pre-immobilized SAV on the T line. Meanwhile, excess free Anti−FITC−GNP will finally be retained by the goat anti−mouse antibody on the C line of the LFS. On the contrary, without target floral origin, it will not induce the production of dual-labeled amplicons. Additionally, the loading of amplification mixture will also not induce the retention of Anti−FITC−GNP conjugates on the T line and the coloration of the T line of the LFS. With this designed genetic authentication strategy, with the presence of target floral origin components in the honey, the target amplicons can be produced and there will be both T line and C line on the molecular LFS. On the contrary, if the honey is entirely fake or partially adulterated with other sweet components, there will be no T line or reduced intensity on the T line while the C line will remain unchanged, which indicates the validation of each test of the LFS. Moreover, the complete genetic authentication of the honey’s floral origin can be performed by normal operators with the portable device in less than 1 h, which will contribute greatly to the guarantee or supervision of food quality, especially honey products. Notably, the primer set designed and adopted in this research is specific for the chloroplast maturase K gene of locust. For the detection of other plant-borne components, only the re-design of primer is required when using this reported method.

### 3.2. Optimization of Rapid Pretreatment of Honey Samples

Traditional isotope-based instrumental standard protocols for honey authentication have strict requirements for the pretreatment of honey samples. Similar protocols with time-consuming and complicated properties cannot meet the requirements of rapid or on-site applications in remote districts. In order to further simplify treatment steps and shorten the duration, a 500 µL viscous honey sample was added into the 700 µL prepared extraction buffer (10 mM Tris−HCl, 200 mM NaCl, 5 mM EDTA, 1% SDS, pH 8.0). The mixture with less viscosity was ultrasonicated fora different duration to destroy the pollen in the sample, and centrifuged at 12,000× *g* for 10 min. Compared with the traditional metal bath method for honey genome DNA extraction, this method can achieve the characteristics of rapidity, simplicity and good repeatability without the need for multi-step operations. As we can see in the Figure 2A, the high purity DNA was obtained from both extraction methods and the DNA quality is adequate for PCR reaction. However, our method for genome DNA extraction was completed in less than 20 min, which greatly improves the detection efficiency. Therefore, this method is particularly suitable for both the fast processing of samples and investigations into the authentication of honey by means of DNA analysis. The extraction efficiency of the different sonication time was considered and compared. From the results in Figure 2B, it can be seen that the extraction efficiency of genetic components from the pollen in the honey increase with the increase in sonication time. The best extraction performance was achieved with an ultrasonication time of 7 min. A further increase in the ultrasonication time will not continue to improve the extraction efficiency, which may be due to the complete destruction of the pollen in the samples or the destruction of the extracted genes. Finally, 7 min was adopted as the best ultrasonication time for the pretreatment.

### 3.3. Optimization of On-Site Amplification and Molecular LFS Detections

To realize the rapid authentication of honey floral origins, other parameters of amplification and the molecular LFS should also be considered. For on-site amplification with the portable thermal controller, the amount of primer and anneal temperature were optimized to achieve suitable results for the LFS. Firstly, for on-site amplification, the primer concentration and the anneal temperature were systematically considered for the better performance. Results in Figure 3A,B demonstrated that 0.16 μM of the primer set and annealed at 55 °C can achieve the better amplification performance. Followingly, for the LFS measurement, the amount of antibody for conjugation on the GNPs were studied. The results in Figure 3C show that 3 μL 1 mg/mL antibody can lead to the strongest visual signal on test lines, which is adopted for later research. The binding of gold nanoparticles to protein usually occurs via the electrostatic adsorption and hydrophobicity, which is based on the relationship between the protein isoelectric point and the pH value of GNPs. The pH value of GNPs is modified by K_2_CO_3_. Under normal conditions, the pH value of GNPs should be modified close to, or slightly higher than, the isoelectric point of protein, so that the adsorption is the most stable and the adsorption efficiency is the highest. Ac−cording to the results shown in Figure 3D, the addition of 5 μL K_2_CO_3_ can contribute to the most distinguishable signal on the test line, indicating the better conjugation efficiency of the antibody on GNPs. The stability of the final suspension system of GNP-Ab conjugates will directly affect the final detection performance. Different proteins were adopted as the stabilizer of the suspension solution. Evidently, the BSA-assisted solution gave the best optical signal on the test line of the LFS as shown in Figure 3E. Finally, for the measurement of amplicons with the molecular LFS, the loading buffer, which provides the migration force on the LFS, was optimized. Five different loading buffers with different components were verified and compared, as depicted in Figure 3F. In detail, the PBS, acting as the loading buffer, can give the strongest signal on test line and the cleanest background, which can also benefit the signal-to-noise treatment. 

### 3.4. Verification of the Practical Performance of Rapid Genetic Authentication Strategy with Molecular LFS

To verify the critical properties of the rapid genetic authentication strategy for honey adulteration, other different common honey samples, with different floral origins, were artificially added to the confirmed real Locust honey at different ratios. Following this, all the artificial adulterated samples were rapidly treated as demonstrated in Section 2.2 and detected with the genetic authentication strategy of the molecular LFS. All detection results of the adulterated Locust honey samples at different ratios were demonstrated in Figure 4. In detail, with the increase in adulteration ratio, the Locust-origin component will be decreased, and the signal on the test line will also be decreased. When the honey sample was entirely fake there was no Locust−origin component in the sample. Accordingly, there was no signal on the test line of the 100% adulteration group (the right strip in Figure 4A). The corresponding ImageJ treated quantitative curves were also shown in Figure 4A; when the adulteration ratio was as low as 25%, the distinguishable variation optical intensity was able to be differentiated just by visual observation. Furthermore, the linear calibration curve was also constructed based on the optical intensities on the test lines of different adulteration ratios. The detection limit of adulteration can be as low as 7.3% by the rule of a 10% signal decrease, compared with the blank group (100% real Locust sample of the left strip in Figure 4A). Meanwhile, the specificity of the constructed genetic authentication strategy was also considered. Other common honey samples—including the rape honey, chaste honey, jujube honey and hazelnut honey samples—were adopted as the control groups. From the results shown in Figure 5, it is clear that only the pure Locust group and the mixture with the presence of Locust honey produced an obvious signal on the test line, indicating the existence of the target Locust component in the honey. However, for other control honey samples, there was no optical signal on the test lines of the different LFS. Meanwhile, for accurate evaluation, the honey sample adulterated at different ratios were determined with the LFS continuously for 10 repetitions. Results show that there is no obvious difference in both the linear relationship and the sensitivity among the 10 tests (See detailed results in Appendix A). Moreover, the LFS of the same batch have also been used for the detection of the same sample at different times. Following this, the inter and intra-day variation were calculated to be 6.82% and 4.31%, and were in the acceptable range of lateral flow strip−based methods. These results strongly prove the excellent specificity of this genetic authentication strategy for the authentication of honey samples. 

### 3.5. Practical Application of Genetic Authentication Strategy for Commercial Honey Samples

Lastly, 14 commercial honey samples, purchased from the local supermarket, were identified with this genetic authentication strategy using the molecular LFS. All detection results are shown in Figure 6; it can be observed that there are obvious signals on the test lines of samples 1, 2, 3, 4, 5, 6, 7, 8, 9, 10 and 12, indicating that there are target Locust components in these commercial honey samples. It should also be noted that there are no observable signals on the test lines of samples 11, 13 and 14. The results primarily indicate that these three samples may be related to the adulteration. These on-site screening results provide direct evidence for further official confirmation. To demonstrate a convincing reliability, the same honey samples were also tested by quantitative real time PCR for confirmation. Results in Appendix A are in great accordance with the obtained PLFS results, indicating the effectiveness and potential practical capability in food quality control.

## 4. Conclusions

Although there are some suggested national standards for the authentication of honey adulteration, none of the current standards are able to guarantee the complete accuracy of the detection. Therefore, the development of a rapid screening method as an effective supplemental and alternative protocol of instrumental standards for accurate identification of honey adulteration is of great significance for routine supervision. In this study, the genetic sequence of the pollen in honey samples was adopted as the target analyte, which can be rapid extracted with the optimized pretreatment method. Efficiency was obviously improved from 3 h to 0.5 h. The target template was amplified on-site to produce enormous functional amplicons with the designed primer set. Additionally, the products of on−site amplifications were skillfully integrated with the molecular LFS and the detection results of amplicons could be easily and visually judged. A similar rapid genetic authentication of honey adulteration with the molecular LFS has been rarely reported. This genetic authentication strategy with the molecular LFS has been well-utilized for commercial honey analysis and satisfactory results have been achieved and confirmed with the national instrumental standard protocol. This genetic authentication strategy with the molecular LFS will provide powerful and effective alternative methods for routine supervision and quality control in the food industry. This method can be further improved with an isothermal amplification step, for use at home by mass consumers, or for on-site detection without professional thermal-control hardware.

## Figures and Tables

**Figure 1 biosensors-12-00971-f001:**
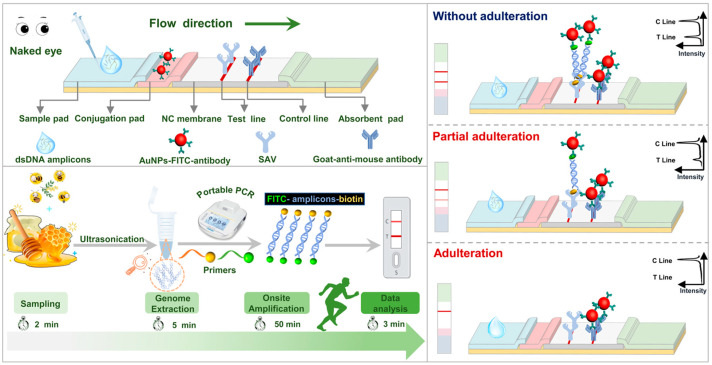
The schematic diagram of the rapid genetic authentication strategy for honey. The target DNA is first extracted by ultrasonication and amplified to produce the enormous functional amplicons with the designed primer set. The products of on-site amplifications were skillfully integrated with the molecular LFS and the detection results of amplicons can be easily and visually judged.

**Figure 2 biosensors-12-00971-f002:**
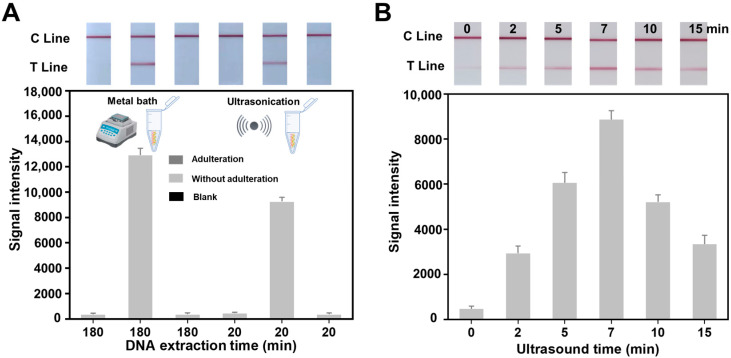
The optimization results of ultrasonic time for pretreatment of honey samples: (**A**) The traditional metal bath method compared with the ultrasonication method for honey genome DNA extraction.; (**B**) Shows the DNA extraction efficiency after different ultrasonication time (0 min, 2 min, 5 min, 7 min, 10 min, and 15 min).

**Figure 3 biosensors-12-00971-f003:**
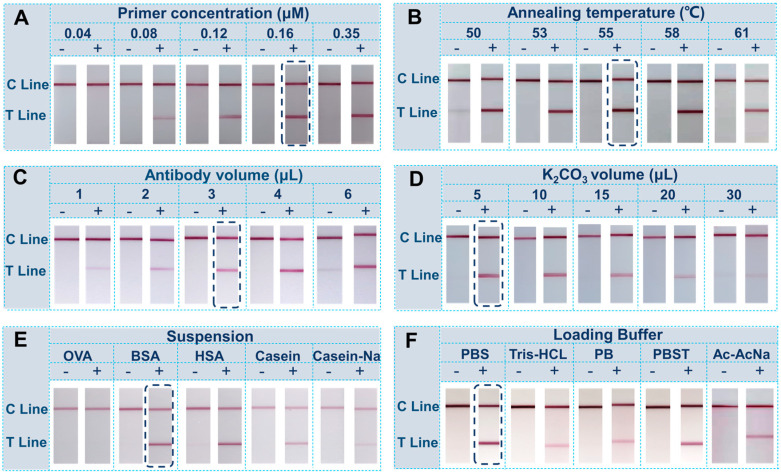
Optimization of experimental parameters: (**A**) Primer concentration (0.04 μM, 0.08 μM, 0.12 μM, 0.16 μM and 0.35 μM); (**B**) Annealing temperature (50 °C, 53 °C, 55 °C, 58 °C and 61 °C); (**C**) Volume of antibody for conjugation (1 μL, 2 μL, 3 μL, 4 μL and 5 μL 1 mg/mL antibody); (**D**) Volume of K_2_CO_3_ for pH adjustment (5 μL, 10 μL, 15 μL, 20 μL and 30 μL 0.1M K_2_CO_3_); (**E**) Suspension species (0.5%OVA, 10% BSA, 10% HSA, 0.5% Casein and 0.5% Casein−Na); (**F**) Categories of loading buffer (10 mM PBS, 10 mM Tris−HCL, 10 mM PB, 10 mM PBT and 10 mM Ac−AcNa).

**Figure 4 biosensors-12-00971-f004:**
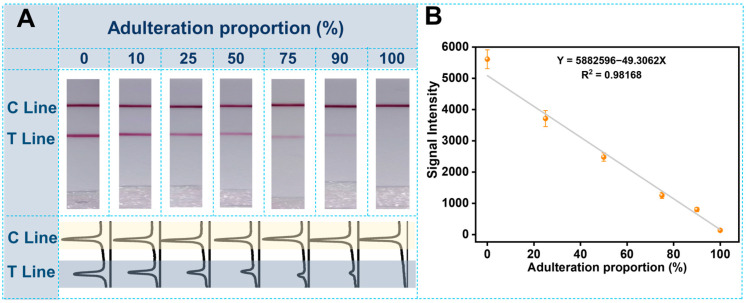
Authentication performance of the amplification−assisted molecular LFS: (**A**) Visual judgment results of honey samples adulterated at different ratio (0%, 10%, 25%, 50%, 75%, 90% and 100%, *v*/*v*%) and the corresponding quantitative curves analyzed by ImageJ; (**B**) Calibration curve of the authentication performance.

**Figure 5 biosensors-12-00971-f005:**
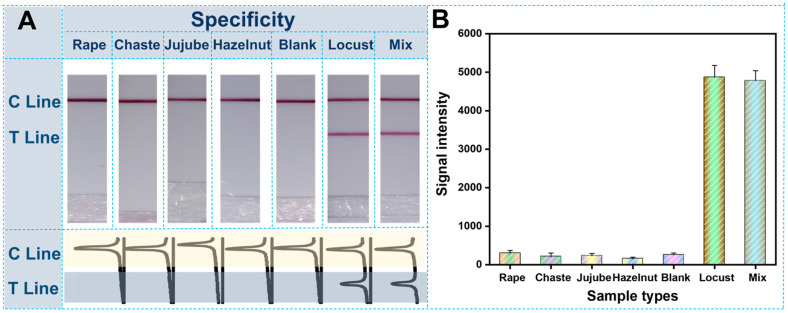
Selectivity assay results of the amplification-assisted molecular LFS: (**A**) Visual observation results of amplification-assisted molecular LFS and corresponding curves of ImageJ treated results; (**B**) Quantitative analysis results of ImageJ treated results.

**Figure 6 biosensors-12-00971-f006:**
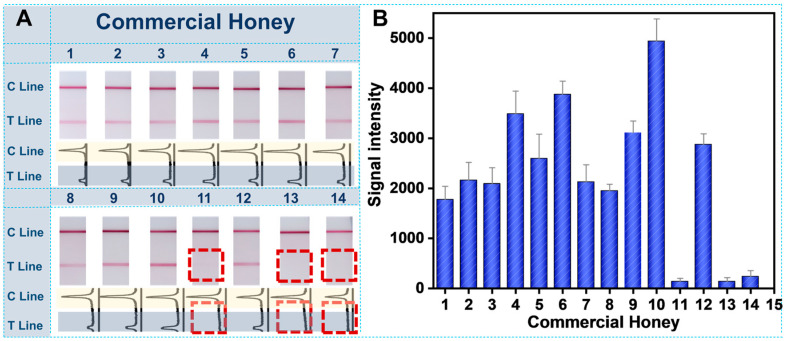
Determination results of 14 commercial honey samples: (**A**) Visual analysis results and corresponding quantitative curves treated by ImageJ; (**B**) Quantitative analysis results by ImageJ.

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
