# Peer review of "Accurate and Rapid Genetic Tracing the Authenticity of Floral Originated Honey with the Molecular Lateral Flow Strip"

_biosensors, 2022, doi:10.3390/bios12110971_

Round 1

Reviewer 1 Report

This manuscript reported the study about honey origin authentication with the molecular method. 

Interestingly, the study not only focused on the development of genetic method for identification of different target genes of flower pollen for honey making, the study also integrated the molecular method with the gold labeled lateral flow strip, then the identification results can be easily judged with visual observation.

The research is significant for honey or food quality control. 

This manuscript can be recommended for publication after the revision.

1 Actually, in the practical applications, for honey samples, there are many floral origins besides the Locust and Rape flowers.

For the accurate screening of the different origins, how to resolve the bottleneck, the authors should make some discussions.

2 The authors just adopted the classic PCR as the molecular protocol. Is there any further improvement about the reported research in this manuscript?

Are there any problems that limit the practical applications of this research?

The authors CAN make some descriptions and discussions. 

Author Response

1 Actually, in the practical applications, for honey samples, there are many floral origins besides the Locust and Rape flowers.

For the accurate screening of the different origins, how to resolve the bottleneck, the authors should make some discussions.

Reply: Thanks a lot for the suggestion. In this study, we designed primers to specifically detect locust components based on the locust chloroplast maturase K gene, which can only amplify DNA containing locust components in the target site. In addition, we can design specific primers for specific DNA fragments for tracing the authenticity of different floral originated honey. Then, for the detection of the potential adulterated components, the specific amplified products can be measured and judged with the lateral flow strip. Based on the comments, we have provided some additional description in the discussion part.

Revisions were labeled in red for review.

2 The authors just adopted the classic PCR as the molecular protocol. Is there any further improvement about the reported research in this manuscript?

Are there any problems that limit the practical applications of this research?

The authors CAN make some descriptions and discussions.

Reply: Thanks for the suggestion. We just adopted the classic PCR as the molecular amplification protocol. In the future, the reported research in this manuscript could be further improved by adopting the isothermal amplification as the molecular amplification protocol to shorten the detection time and simplify the dependence of the thermal-control hardware.

We have provided some additional description and discussions in the revised manuscript and all revisions were labeled in red for review.

Reviewer 2 Report

The present study has scientific merit for detection of authentic honey, however, the authors need to describe clearly the working principle behind the LFS assay. I had difficult time to understand the underlying principle. How genetic sequence of pollen could be used for such analysis? Will it not dependent on plant species? Can same pollen sequence be used for any plant source?

Apart from this concern, authors need to elaborate captions for Figure 1, 2, and 3. There is very little information provided for this figures in these captions. The design of assay and outcomes should be described in details.

Remaining of the manuscript text is well-written and aptly presented. 

Author Response

The present study has scientific merit for detection of authentic honey, however, the authors need to describe clearly the working principle behind the LFS assay. I had difficult time to understand the underlying principle. How genetic sequence of pollen could be used for such analysis? Will it not dependent on plant species? Can same pollen sequence be used for any plant source?

Apart from this concern, authors need to elaborate captions for Figure 1, 2, and 3. There is very little information provided for this figures in these captions. The design of assay and outcomes should be described in details.

Remaining of the manuscript text is well-written and aptly presented.

Reply: Thanks a lot for the patient comments. In order to identify the origin of the honey, the characteristic marker should be carefully selected. Many targets can be adopted as the marker for authentication research. Considering the production process of honey, the existence of pollen in the honey could be adopted as the target marker for authentication research, which is strictly and highly related to the authenticity of the honey. The specific pollen sequence can just be used for the corresponding plant. For the judgement of other plant source, the specific pollen sequence of the plant should be analyzed and corresponding primer set should be designed for analysis. Therefore, the sequence of locust chloroplast maturase K gene can not be used for other plant source analysis. Based on the comment, we have added some supplemental description about adoption of pollen as the target.

To be more clearly show the detection principle, the captions of mentioned figures have been further described in detail and the detailed working principle of the LFS assay was added on Line 158-165 of the revised manuscript.

All revisions were labeled in red fore review in the revised manuscript.

Reviewer 3 Report

This manuscript presents, in a nutshell, the identification of adulterated honey and classification based on 'quality', using a lateral flow strip. Even when the topic seems of interest in the agricultural arena, the use of lateral flow immunoassays are widely spread and quite established. Therefore, the presented work has a low impact in terms of novelty in the context of biosensing (i.e. Biosensors journal).  Modification of the functionalization of the NPs in the strips is something that is achieved in a straightforward fashion, and the results presented are limited to a comparison of commercial honeys and adulterated ones using the strips. The results are based on binary readings that are later translated to a characterization based on the intensity of the (positive) line on the strip. The manuscript also has several flaws in terms of grammar and style that must be cleared/improved. For these reasons, I consider that the manuscript is not ready for publication in Biosensors in its current state.

Author Response

This manuscript presents, in a nutshell, the identification of adulterated honey and classification based on 'quality', using a lateral flow strip. Even when the topic seems of interest in the agricultural arena, the use of lateral flow immunoassays are widely spread and quite established. Therefore, the presented work has a low impact in terms of novelty in the context of biosensing (i.e. Biosensors journal). Modification of the functionalization of the NPs in the strips is something that is achieved in a straightforward fashion, and the results presented are limited to a comparison of commercial honeys and adulterated ones using the strips. The results are based on binary readings that are later translated to a characterization based on the intensity of the (positive) line on the strip. The manuscript also has several flaws in terms of grammar and style that must be cleared/improved. For these reasons, I consider that the manuscript is not ready for publication in Biosensors in its current state.

Reply: Many thanks for the kind suggestions. Based on the suggestions, we have made further improvement of the manuscript. For the method we adopted in this research, it is of little difference from the traditional lateral flow immunoassays. Traditional lateral flow assay focused on the direct recognition and detection of target analytes by immunorecognition principle in the sandwich or competitive models. In this research, the target for latera flow assay is the double stranded DNA obtained by pre-amplification. In this way, the high amplification efficiency of PCR and the convenience were well integrated for the authenticity identification. This is what we want to highlight in this research. For adulteration identification of honey, the current method is based on the isotope-based instrumental methods. And for rapid screen, we just make a try using this amplification integrated lateral flow strip in this field. And as the reviewer mentioned, finally, just the binary readings can realize the results judgements.

Based on the suggestion, we have made further discussions in the manuscript to highlight the research in this manuscript.

Besides, the whole manuscript has been carefully checked to correct the grammar and style errors.

All revisions have been labeled in red for review.

Round 2

Reviewer 2 Report

No comments.

Reviewer 3 Report

The manuscript has undergone some minor changes. Therefore, it is not suitable for publication in Biosensors in its present form.